# Public Space Satisfaction Evaluation of New Centralized Communities in Urban Fringe Areas—A Study of Suzhou, China

**DOI:** 10.3390/ijerph20010753

**Published:** 2022-12-31

**Authors:** Yukang Song, Yong Wang, Min Zhou

**Affiliations:** 1School of Architecture and Urban Planning, Suzhou University of Science and Technology, Suzhou 215000, China; 2Jiangsu Province Planning and Design Groups of Suzhou Branch, Suzhou 215000, China

**Keywords:** public space satisfaction, urban fringe area, new centralized community, grey correlation analysis, Suzhou

## Abstract

The fast and pronounced changes in dwelling space in urban fringe areas, caused by rapid urbanization, has led to the appearance of new centralized communities. These communities possess characteristics of both urban and rural communities and have been facing great pressure related to the reconstruction of social network relations and the cultivation of a community identity. The outcomes of public space satisfaction evaluations are related to the social functions they fulfill, such as motivating social interaction, cultivating community identity, and integrating social relationships. This study evaluates public space satisfaction based on the study of six new centralized communities in an urban fringe area of Suzhou, using grey relational analysis. The results show that the overall satisfaction value is a standard level. Moreover, public space satisfaction is related to patterns of centralization and factors of social culture; especially the latter has great influence on public space satisfaction. Factors related to public space satisfaction can generally be divided into quadrants of “Low Satisfaction–High Significance” and “High Satisfaction–Low Significance.” According to the inverse correlation between satisfaction level with public space and significance of indexes, we propose that indexes of public space satisfaction in the “Low Satisfaction–High Significance” quadrant should be improved greatly during the process of the optimization and construction of public spaces in new centralized communities.

## 1. Introduction

With rapid urbanization in many countries, rural areas have been integrating into urban areas at an accelerating pace. Traditional settlement patterns of urban fringe areas are being gradually assimilated by urban landscape, and they are beginning to collapse and divide. New centralized communities, as a kind of special and new form of dwelling or residence, are starting to appear. Recently, researchers have not been able to reach an agreement on the definition of “new centralized community”. Wang Yong pointed out that under the influence of the disordered construction of villages and the shortage of construction land use, the new centralized community is formed by the centralization of scattering villagers [1]. Xiang Chuang pointed out that new centralized communities form through the transaction and transformation of land development rights due to related national policies [2]. This paper proposes that new centralized communities should be planned and constructed cohesively by the government for those villagers who lost their land and livelihood under the condition of urban sprawl and the development of new residential areas. This type of community is an environment that serves as a transit space from traditional settlements to urban communities. Compared with traditional settlements, new centralized communities not only inherit social networks, culture, and customs, they also initiate changes in human settlement environments by improving public facilities. Compared with urban communities, the sudden and pronounced changes in dwelling space and the influx of migrant workers attracted by rent advantages, new centralized communities face various social problems, such as the inhabitants’ demands being overlooked, the deconstruction of living communities, broken social networks, and a decline in community identity. As the mature urban communities of the future, answering the question of how to satisfy inhabitants’ demands and reconstruct living areas to promote the stable and harmonious development of new communities has become the key to realizing the benign development of cities. Moreover, urban–rural integration must also be addressed urgently in academia. As a public communication medium, public space plays an important role in various social functions, such as promoting the integration of social relationships [3,4], enhancing a sense of community identity [5,6], inheriting collective memories [7], and strengthening community cohesion [8]. Inhabitants’ level of satisfaction with public space is closely related to the social functions of the public space, such as satisfying their demands, motivating social interaction, and building community identity. Satisfaction with a public space is the standard of how satisfied residents are with the environment, physically and mentally. It is therefore of great significance to the successful transition of new centralized communities to explore methods of building public spaces in new centralized communities to satisfy residents’ demands and develop a community identity in urban fringe areas.

The factor of level of satisfaction originated in the field of marketing management as a measure of coherence between expectations and reality [9]. Since the 1950s, research on satisfaction has expanded into other fields, such as urban planning and geography. In terms of research content, the research focuses on the connotations of public space satisfaction [10,11], influencing factors [12,13], evaluation model construction and measurement [14,15], etc. Existing research suggests that “Service Access”, “Social Security”, “Dwelling Record”, and “Physical Specifications of Dwelling Place” have a significant relationship with “Youth Satisfaction” [13]. In terms of research objects, urban public spaces [14,16], such as parks [17], squares [18], and greenways [19], are our main study objects. Other studies have focused on surveys of satisfaction with the quality of life of informal communities in urban fringe areas, showing that components of general dissatisfaction with the quality of life are related to transportation, leisure, basic services (water, energy, etc.), governance, etc. [20]. High-quality public space can improve residents’ sense of desire and happiness index [21]. However, there are relatively few research results devoted to the satisfaction of public space at the micro-community level in urban fringe, and most of them focus on the social level [22,23,24], which does not involve physical space very often. This study applies the grey relational analysis (GRA) method [25] to analyze the levels and factors of public space satisfaction of new centralized communities in urban fringe areas, based on the study of six new centralized communities in Suzhou, China, to determine the key factors that influence public space satisfaction. We expect this paper to be helpful in guiding the optimization and construction of public space as well as promoting both a sense of community identity and the transformation of communities.

## 2. Method and Objects of Research

In this paper, grey correlation analysis is selected as the satisfaction evaluation method for a new concentrated community public space. Six communities with obvious concentrated characteristics are selected as the research objects in the marginal areas of Suzhou. Interviews, questionnaires, field surveys, and other means of community departments are adopted to evaluate the satisfaction of new concentrated community public space in the marginal areas of Suzhou.

### 2.1. Method of Research

Currently, regression analyses, standard deviations, and factor analyses are the methods that are mainly applied to research public space satisfaction. A regression analysis is suitable for use in a space satisfaction analysis in which the sample distribution obeys the probability characteristic distribution [26,27]. The standard deviation method is mostly applied to evaluate public space satisfaction on a physical level, and it contains subjective randomness [28]. The factor analysis method is often used to induce certain types of factors of public space satisfaction [29]. The factors of public space satisfaction comprise not only physical elements, such as symbols and forms, but also social elements, such as activities and management. These factors are too complex to quantify and judge subjectively. The GRA method is suitable for quantifying these types of objects; it can not only reveal relationships among factors that influence public space satisfaction but also be concurrent with the fuzzy characteristics of factors, whether the samples have probability characteristics or not [25]. Therefore, GRA is superior to the other methods mentioned above for evaluating the public space satisfaction of new centralized communities. 

### 2.2. Objects of Research

Suzhou is a regional megacity in China. Since its implementation of the policy of Three Concentrations in the late 1990s, many new centralized communities have appeared on the urban fringe of Suzhou. According to the Suzhou Bureau of Land and Resources, 680 new centralized communities had been built by the end of 2016. These communities include 727,000 central resettlement families; moreover, the centralization rate of residents is up to 53.2%. The new centralized community has become the most sensitive and active unit during the process of urban–rural integration. Considering the differences among different cities’ topographic features, production areas, and lifestyles, each area adopts a construction pattern according to its own characteristics during the development of new centralized communities. For example, Gan Xinkui divides communities into three construction patterns according to levels of economic development. These levels form patterns against the background of the urban sprawl, the in situ urbanization (villagers changing their rural lifestyles into urban life by developing the economy locally without migrating to urban areas), and the self-governance of villagers [30]. Based on their investigation in Hubei Province, Xu Yuanwang states that communities can be divided into types according to community organization boundaries; these are town type, established village type, natural village type, and central village type [31].

Based on a study of communities in the urban fringe areas of Suzhou, this study divides new centralized communities into two types: heterogeneous centralization and local centralization, according to their space scale and causes of formation. The former community type was mainly constructed under the guidance of the government during the period when development zones and industry parks were built. They usually feature massive concentration, early ages, and multi-story buildings. The latter community type was mainly formed during a period of village collectives reviving construction land. They usually feature small-scale concentration, new ages, and multi-story and small high-rise buildings. Considering the integrity of the region, as well as the accuracy and scientificity of the research results, we selected the objects to survey according to some contrastive elements, such as community size, centralization pattern [32], and construction age. Based on these requirements, investigations, and interviews with the departments of communities in Suzhou urban fringe areas, we selected six typical communities as the final research objects, including Anyuan Community, Jinyun Community, Huifeng Community, and Lianhua Community (Table 1).

## 3. Research Design and Data Processing

Satisfaction with public space is affected not only by subjective factors such as residents’ age and education level, but also by visitor factors such as space environment, hardware facilities, and management level, which is a two-way interaction process between residents and places. This paper constructs a conceptual model of public space satisfaction in new-type concentrated communities, selects six typical new-type concentrated communities on the edge of Suzhou as empirical research objects, and uses grey correlation analysis to evaluate their public space satisfaction.

### 3.1. Construction of Evaluation Index System

Public space satisfaction levels are influenced not only by the activity perception of residents, but also by the environment, facilities, and management, for example. Several scholars have been studying this; for example, Winter constructed the index for community public space satisfaction in terms of landscape environment, facilities, maintenance, and management [33]. Rapoport considers that public space can be used more often by holding traditional celebrations regularly [34]. Ye Jihong holds the view that facilities are key factors that influence public space, and that the appropriate distribution of facilities can promote neighborhood communication. Xie Honghai thinks that the effective operation of public space can advance the construction of residents’ social networks [35], while Yuan Yuan points out that setting cultural landmarks in public spaces can help promote the construction of community identity [6]. Based on this, this paper summarizes the satisfaction evaluation index of public space into four categories: satisfaction with the space environment, satisfaction with social management, satisfaction with site facilities, and satisfaction with the cultural environment. We invited 30 experts (including community administrators, community planners, and professors) and 80 community residents to participate in the selection of evaluation indexes and established a new centralized community public space satisfaction sample database based on formal questionnaires and field interviews. SPSS 21.0 software was used for reliability, validity, and exploratory factor analysis, and indicators with Cronbach’s α less than 0.7 and validity (KMO and Bartlett sphericity test) less than 0.5 were eliminated [36], such as pavement form, functional layout, and perfection of pointing sign. Finally, from both material and social aspects, the satisfaction evaluation index system of the new centralized community public space in the urban fringe is determined, which includes 15 indexes in four dimensions, namely space environment, site facilities, social management, and cultural environment (Table 2).

### 3.2. Data Source and Sample Description

First, a survey questionnaire on public space satisfaction was designed based on the objects being surveyed and the index system. The first part of the questionnaire recorded the participants’ characteristics, while the second part quantified the evaluation indexes. The Li Kete scale method was applied in the questionnaire [37], with 1 representing the least satisfaction and 5 representing the most satisfaction. Using a random sampling method, one family was selected from each building to participate in the research from September 2016 to December 2016. The group distributed 500 questionnaires, and 443 valid questionnaires were returned, a response rate of 88.6%. As shown in Table 3, among these interviewees, 26.4% were older people, aged older than 61; this is over the average age of Suzhou City. The household income per capita was approximately 1/3 that of the average in Suzhou City. The average education level was a little low, with 60% of the interviewees’ education levels being lower than an associate degree. 59.2% of interviewees make a living as servants, workers, or self-employed households, showing that employment levels were generally low.

Second, the reliability of the 443 valid questionnaires was tested using SPSS v 22.0 software. The results show a Cronbach’s α coefficient of 0.786, higher than the least acceptable number, 0.6 [17]; this confirms the reliability and stability of the questionnaire.

### 3.3. Data Processing of Public Space Satisfaction Level

The core soul of public space in suburban concentrated communities is to serve residents, and the objective differences in environment, facilities, management, and other aspects will affect residents’ satisfaction with public space. Using grey correlation analysis, it was found that all objective factors are related to public space satisfaction, but there are significant differences in the influencing factors and degrees of public space satisfaction in different communities. 

#### 3.3.1. Determination of Reference Sequence and Nondimensionalization

GRA was used to determine the reference sequence and comparison sequence of public space satisfaction levels of new centralized communities based on qualitative analysis (Table 4). To ensure the reliability of the analysis results, an initial value method was applied to nondimensionalize the questionnaire data.

#### 3.3.2. Weight Calculation of Evaluation Indexes

An analytic hierarchy process was applied to determine the index weight due to the difference in indexes’ influence on public space satisfaction. We invited 30 experts (including community administrators, community planners, and professors) and 80 residents to participate in the comparison and judgment of indexes. The weight result is shown in Table 5, after data processing using yaahp10.6 software. The number of consistency checks ranges from 0 to 0.026, still below 0.1, indicating that it can be used as evaluation index weight for public space satisfaction of new centralized communities. For example, *W*_k_ is the weight of the evaluation index No. k in Formula (1).
(1)$$W=\left\{W_{k}|k=1,2,3,\cdots ,n\right\}$$

#### 3.3.3. Grey Weighted Relational Grade Calculation

Grey weighted relational grade is the measurement that indicates the correlation strength between public space satisfaction and evaluation indexes. The higher the grade, the more satisfied residents feel with community public spaces, and the more influence there was on satisfaction by evaluation indexes. Furthermore, it is a distinguishing coefficient that always takes the value of 0.5 [30], and it is a difference sequence, meaning that it is the absolute value difference between the comparison sequence curve and the reference sequence curve at the point of k after the non-dimensionalization of the questionnaire data on public space satisfaction. M and m are the maximal value and minimum value among absolute differences, respectively.
(2)$$\gamma_{0i}=\sum_{k=1}^{n}W_{k}\;\;\frac{m+\rho \cdot M}{\Delta_{0i}(k)+\rho \cdot M}(k=1,2,3,\cdots n;i=1,2,3\cdots m)$$

#### 3.3.4. Satisfaction Measurement of Public Space

Next, we determined the average numbers of the evaluation indexes of public space satisfaction as a comparison sequence, and the maximum value of each index as a reference sequence. We then applied Formula (2) to evaluate the public space satisfaction of new centralized communities and obtain the results, shown in Table 6. Table 6 shows that the maximum difference is 50, and the minimum difference is 0. This result indicates that the grey weighted relational grade of public space satisfaction of new centralized communities is 0.584, among which the average grade of heterogeneous centralization type is 0.577, while the average grade of local centralization type is 0.591.

#### 3.3.5. Index Evaluation of Public Space Satisfaction

We made a list of average numbers of evaluation indexes of public space satisfaction as a comparison sequence, and took the maximum numbers of indexes as the reference sequence. We then applied Formula (2) to evaluate the grey weighted relational grade of public space satisfaction evaluation indexes, as shown in Table 7, and sorted numbers by the number magnitude to determine the leading factors that greatly influence public space satisfaction. In the equation, M is 50, and m is 0. According to the order of grey weighted relational grade of the evaluation index, we can conclude below:

## 4. Result of the Analysis of Public Space Satisfaction of New Centralized Communities

The results show that the value of public space satisfaction of new centralized communities is 0.584, which is an average level (0–0.2: very unsatisfied; 0.2–0.4: unsatisfied; 0.4–0.6: average level; 0.6–0.8: satisfied; 0.8–1.0: very satisfied). From this result, we can conclude that the overall level of public space satisfaction in new centralized communities is not high, indicating that although many opportunities are provided to improve the quality of public spaces due to the construction of new centralized communities, many problems still remain. This lowers the overall level of public space satisfaction. Somehow, during the construction of public spaces and distribution of facilities, residents’ demands for social culture, social relationships, public activities, and cultivating a sense of identity are overlooked. This makes residents feel downhearted and indifferent to the new community.

### 4.1. Degrees of Influence of Different Centralization Patterns on Public Space Satisfaction

By comparing the results of public space satisfaction of the two types of new centralized communities, it can be concluded that the level of public space satisfaction is related to the centralization patterns of communities. The satisfaction level of local centralization-type communities is higher than those of heterogeneous centralization-type communities, with values of 0.591 and 0.577, respectively.

Compared with local centralization-type communities, heterogeneous centralization-type communities were always constructed earlier, with lower environment quality and older facilities. Most local centralization-type communities were constructed within the last seven years, with relatively better facilities. Furthermore, heterogeneous centralization-type communities were used to develop due to the cross-administration of villages, and properties were allotted by lot. Traditional acquaintance society has transformed into a semi-acquaintance society, or even stranger society, which is damaging to social communication between residents of the community. Furthermore, public activities became more individual, with the community identity remaining to be built up in heterogeneous centralization-type communities. A modern governance mode is applied to the governance of local centralization-type communities, while community organizations are relatively mature with a higher frequency of public activities and traditional celebrations that residents are willing to participate in. In addition, local centralization-type communities, where villagers have similar backgrounds, used to concentrate within villages with good conservation of the former social networks and emotional foundation. Therefore, the level of public space satisfaction of local centralization communities is a little higher than that of heterogeneous centralization-type communities.

### 4.2. Social Environment Is the Key Factor That Influences Space Satisfaction Level

Based on the analysis of the key factors that influence the public space satisfaction of new centralized communities, we conclude that the evaluation indexes above are related to the level of public space satisfaction, while there are differences in the degree of influence on it (Table 7). The indexes of activity participation, activity diversity, and activity frequency greatly influence public space satisfaction, with values of 0.249, 0.224, and 0.219, respectively. Indexes such as perfection of lighting installation, perfection of sanitation facilities, and brick rate, however, have a relatively smaller influence on public space satisfaction, with values of 0.165, 0.154, and 0.153, respectively. In summary, social environment factors have a much greater influence on the level of public space satisfaction of new centralized communities compared with that of physical space.

There are two reasons for this. First, social relationships, social capital, and community identity have changed a lot since the centralization of communities. Offering various activities for residents to participate in regularly in community public spaces can not only help build close relationships between residents, but also help residents to share their sense of value and local customs with each other. This, in turn, may promote the development of community identity. In this sense, social environment factors have a great influence on the level of public space satisfaction. Second, although the new centralized community is constructed according to high standards, its management institutions must still be transformed. Sustainable development of residents’ social communication, relaxation, and community activities must all be guaranteed through effective management and maintenance in public spaces. In this sense, social management has a great influence on the level of public space satisfaction. Therefore, during the construction of new centralized communities, not only the perfection of physical space should be addressed, but also the social environment, with an emphasis on factors such as public activities, participation of residents, and the management institution of communities.

### 4.3. Inverse Correlation between Level of Space Satisfaction and Significance of Indexes

Next, we address the degree of influence of indexes on public space satisfaction as the significance of indexes (Y axis) and levels of public space satisfaction on the X axis. We analyzed the correlation between levels of satisfaction (X axis) and significance (Y axis) to deduce four quadrants according to the value average of significance and satisfaction levels (Figure 1).

The results show that the three indexes of accessibility, daylighting, and perfection of fitness facilities were located in the “High Satisfaction–High Significance” quadrant. Activity participatory, activity diversity, activity frequency, cultural landmark richness, management, and maintenance of facilities and environment were concentrated in the “Low Satisfaction–High Significance” quadrant. The perfection of resting facilities, vegetation richness, safety, perfection of lighting facilities, perfection of lighting installation, and brick rate are in the “High Satisfaction–Low Significance” quadrant. From the perspective of the factor distribution of satisfaction in public space, most factors are concentrated in the “Low Satisfaction–High Significance” and “High Satisfaction–Low Significance” quadrants, while few factors are distributed in the “High Satisfaction–High Significance” quadrant, showing a significant reverse relationship. This also shows that social and cultural factors that are more important to residents’ daily life, such as participation, diversity, and frequency of activities. The lack of corresponding attention in the construction and management of new centralized communities results in low satisfaction with public space. However, the relatively low level of importance of the brick rate, the perfection of sanitation facilities, lighting installation, etc., has a high degree of satisfaction, indicating that the construction of a new centralized community public space at the material level can basically meet the needs of residents at the current stage of life.

## 5. Conclusions

### 5.1. Conclusion

The public spaces of new centralized communities in urban fringe areas are the vessel of residents’ daily life, where residents can have social communication and public participation. It is also the junction between inheriting community spirit and bearing social relationships. Research on the public space satisfaction of new centralized communities is of great significance, especially in terms of its influence on the sustainable development of residents’ activities, the reconstruction of social networks, and the cultivation of community identity [31,38]. This study chose six typical new centralized communities in the Suzhou fringe area as research objects because Suzhou plays a leading role in the process of urbanization in China. Based on our analysis of public space satisfaction evaluation of new centralized communities, we conclude the following:(1)The overall satisfaction level of new centralized communities’ public spaces is 0.584, which is at an average level; there is still some way to go to reach the level of “very satisfied.” The construction of new centralized communities should emphasize not only the creation of high-quality physical public spaces but also the cultivation of social relationships and community identity to improve public space satisfaction levels.(2)When comparing public space satisfaction levels of heterogeneous centralization-type communities and local centralization communities, the results showed that different patterns of centralization are prerequisites that influence public space satisfaction. Different patterns of centralization are related to aspects of environment quality, perfection of facilities, degree of social relationship breakage, and community management institutions. By affecting the aspects above, centralization patterns influence the results of public space satisfaction.(3)Analyzing the leading factors that influence public space satisfaction levels, we discovered that factors of social environment have a much greater impact on the overall public space satisfaction of new centralized communities than those related to the physical environment. This indicates that social attributes are much more important than physical attributes when it comes to public space satisfaction.(4)By analyzing the correlation between levels of public space satisfaction and significance of indexes, we showed that there is an inverse correlation between satisfaction level and significance, with most of the indexes in the quadrants of “Low Satisfaction–High Significance” and “High Satisfaction–Low Significance.”

### 5.2. Suggestions

Existing research on public space satisfaction has rarely considered communities at the micro level before. This study eliminates the traditional method of evaluating public space satisfaction in terms of material based on existing studies. We chose six typical new centralized communities in the Suzhou fringe area as objects to perform empirical research from the perspective of social space. Although the reliability of the results may be influenced by the small sample, this study provides a new perspective for future research on the construction of similar new centralized communities and the improvement of public space satisfaction levels. According to the correlations between the levels of public space satisfaction and the significance of indexes, suggestions for the optimization and construction of differential public spaces are proposed as follows.

(1) Keep indexes in the quadrant of “High Satisfaction–High Significance” as they are, including accessibility, daylighting, and perfection of fitness facilities. The advantages offered by these indexes should be maintained and improved. For example, the perfection of fitness facilities can help cultivate the vitality of public spaces, which is beneficial for social relationships. This can also inspire individual or social activities to promote residents’ communication. It is necessary to equip various fitness facilities and places to adapt to the demands of residents of different ages.

(2) The promotion of indexes in the quadrant “Low Satisfaction–High Significance”, such as activity participation and activity diversity. These indexes are key factors that influence residents’ public communication, the cultivation of community identity, and the transformation of communities. Promoting these indexes will be highly beneficial for the promotion of public space satisfaction levels. The following aspects must be promoted urgently: First, propaganda and education should be conducted to raise residents’ awareness of and ability to participate in public activities and community affairs. Second, community collective activities, such as traditional celebrations and performances, should be held regularly to create more opportunities for local residents and immigrants to communicate with each other and develop collective memories, pushing forward the development of a community identity. It is also beneficial to maintain the countryside spirit by setting landmarks featuring local characteristics in community public spaces. Finally, property management companies or pluralistic organizations could be introduced by the force of market mechanisms to establish scientific space governance policies to guarantee the functions of public spaces.

(3) Optimize the indexes in the quadrant of “High Satisfaction–Low Significance” gradually, including perfection of resting facilities, vegetation richness, and safety. These indexes play an important role in maintaining satisfaction and attraction in public spaces. However, the marginal profit of these indexes is a little small; while investing the same input compared with other indexes, it is not necessary to invest too much in funds and resources in the promotion of these indexes when there is a lack of funds. It is therefore advised to keep these indexes as they are, just in the case of reducing the overall satisfaction level of public space.

## Figures and Tables

**Figure 1 ijerph-20-00753-f001:**
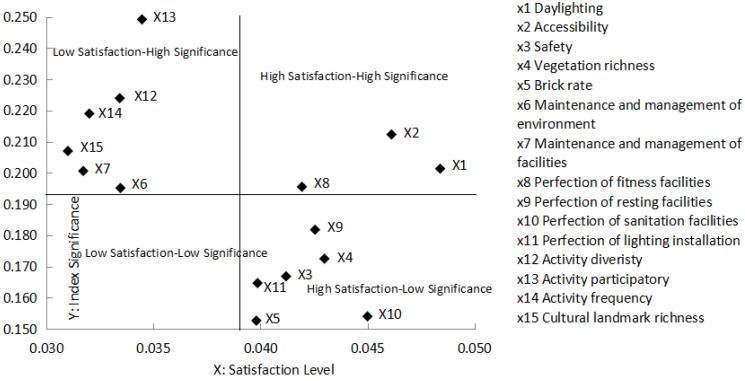
Quadrant distribution of public space satisfaction level–index significance of new centralized community.

**Table 1 ijerph-20-00753-t001:** Introduction of the communities surveyed.

	Huifeng Community	Lianhua Community	Shanhu Community	Anyuan Community	Jinyun Community	Jinsongwan Community
Centralization Pattern	Heterogeneous centralization			Local centralization		
Location	Suzhou New District	Suzhou Industrial Park	Wujiang District	Xiangcheng District	Wuzhong District	Suzhou Industrial Park
Size	28.3 hm^2^	32.3 hm^2^	30.1 hm^2^	7.5 hm^2^	19.8 hm^2^	12.6 hm^2^
Construction Age	2005	2005	2005	2010	2010	2010

**Table 2 ijerph-20-00753-t002:** Attribute characteristics of public space satisfaction evaluation indexes of a new centralized community.

First-Level Index	Second-Level Index	Attribute of Index	First-Level Index	Second-Level Index	Attribute of Index
Satisfaction of space environment	Daylighting X_1_	Satisfaction of insolation duration	Satisfaction of site facilities	Perfection of fitness facilities X_8_	Satisfaction of number and distribution of fitness, resting, lighting and health facilities
	Accessibility X_2_	Satisfaction of convenience of arrival		Perfection of resting facilities X_9_	
	Safety X_3_	Satisfaction of security and walking environment safety		Perfection of sanitation facilities X_10_	
	Vegetation richness X_4_	Satisfaction of plant species		Perfection of lighting installation X_11_	
	Brick rate X_5_	Satisfaction of ratio of brick area to total area	Satisfaction of cultural environment	Activity diversity X_12_	Types of traditional and folk activities
Satisfaction of social management	Maintenance and management of environment X_6_	Satisfaction of management of health and green		Activity participatory X_13_	Dwellers’ participation in community activities
	Maintenance and management of facilities X_7_	Satisfaction of maintenance of fitness, resting, lighting, health and landmarks facilities		Activity frequency X_14_	Frequency of holding activities regularly
				Cultural landmark richness X_15_	Continuity of traditional settlement collective memory

**Table 3 ijerph-20-00753-t003:** Characteristics of the samples.

Attribute	Type	Sample Number	Percentage (%)	Attribute	Sample Number	Percentage (%)
Sex	Male	185	41.8%	Education level	151	34.0%
	Female	258	58.2%		132	29.8%
Age	Under 18	23	5.1%		87	19.7%
	18~30	107	24.1%		55	12.5%
	31~45	99	22.3%		18	4.0%
	46~60	97	22.1%	Occupation	25	5.6%
	Over 61	117	26.4%		56	12.6%
Household income per capita	Under 1000 yuan	70	15.8%		89	20.2%
	1001–2000 yuan	124	28.1%		108	24.3%
	2001–3000 yuan	131	29.6%		99	22.3%
	Over 3001 yuan	118	26.5%		66	15.0%

**Table 4 ijerph-20-00753-t004:** The score of new centralized community public space satisfaction.

Evaluation Index	X_1_	X_2_	X_3_	X_4_	X_5_	X_6_	X_7_	X_8_	X_9_	X_10_	X_11_	X_12_	X_13_	X_14_	X_15_
Huifeng Community	2.3	2.5	3.4	3.3	3.0	3.1	3.3	2.4	2.8	3.2	2.8	2.3	4.0	3.3	3.3
Lianhua Community	2.7	2.7	3.4	3.3	3.1	2.5	3.0	3.2	2.8	3.2	3.0	2.2	2.5	3.2	3.5
Shanhu Community	3.0	2.4	3.0	3.0	3.2	2.8	2.4	2.3	2.7	3.2	3.3	4.1	2.4	2.6	3.3
Anyuan Community	2.5	2.0	3.4	3.3	3.3	3.0	2.8	2.7	2.5	3.1	2.7	2.2	3.0	3.6	4.0
Jinyun Community	2.5	3.3	3.3	2.3	3.2	2.8	3.8	2.5	3.2	2.8	3.2	2.3	2.7	2.8	3.2
Jinsongwan Community	3.2	3.0	3.3	2.5	3.2	3.8	2.3	2.3	2.5	3.1	3.0	2.0	2.8	4.0	3.0

**Table 5 ijerph-20-00753-t005:** Weight of evaluation index of public space satisfaction of new centralized communities.

Evaluation Index	Combined Weight	Evaluation Index	Combined Weight	Evaluation Weight	Combined Weight
Daylighting X_1_	0.084	Maintenance and management of environment X_6_	0.067	Perfection of lighting installation X_11_	0.058
Accessibility X_2_	0.087	Maintenance and management of facilities X_7_	0.064	Activity diversity X_12_	0.087
Safety X_3_	0.051	Perfection of fitness facilities X_8_	0.082	Activity participatory X_13_	0.062
Vegetation richness X_4_	0.059	Perfection of resting facilities X_9_	0.073	Activity frequency X_14_	0.070
Brick rate X_5_	0.047	Perfection of sanitation facilities X_10_	0.049	Cultural landmark richness X_15_	0.060

**Table 6 ijerph-20-00753-t006:** Grey weighted relational grade of public space satisfaction of new centralized communities.

Pattern	Community	X_1_	X_2_	X_3_	X_4_	X_5_	X_6_	X_7_	X_8_	X_9_	X_10_	X_11_	X_12_	X_13_	X_14_	X_15_	∑
	Huifeng	0.030	0.034	0.047	0.059	0.031	0.031	0.035	0.032	0.041	0.051	0.030	0.023	0.029	0.028	0.070	0.571
Heterogeneous	Lianhua	0.042	0.040	0.047	0.059	0.035	0.021	0.027	0.082	0.041	0.051	0.037	0.022	0.035	0.026	0.021	0.587
centralization	Shanhu	0.060	0.032	0.027	0.037	0.041	0.025	0.019	0.029	0.037	0.051	0.058	0.087	0.029	0.019	0.020	0.572
	Anyuan	0.039	0.028	0.047	0.059	0.049	0.031	0.026	0.045	0.034	0.044	0.029	0.025	0.062	0.039	0.030	0.586
Local centralization	Jinyun	0.035	0.087	0.040	0.020	0.041	0.025	0.064	0.034	0.073	0.028	0.049	0.023	0.027	0.021	0.023	0.591
	Jinsongwan	0.084	0.055	0.040	0.023	0.041	0.067	0.018	0.029	0.031	0.043	0.037	0.021	0.024	0.060	0.024	0.597
Average		0.048	0.046	0.041	0.043	0.040	0.033	0.032	0.042	0.043	0.045	0.040	0.033	0.034	0.032	0.031	0.584

**Table 7 ijerph-20-00753-t007:** Grey weighted relational grade of public space satisfaction indexes of new centralized communities.

Index	X_1_	X_2_	X_3_	X_4_	X_5_	X_6_	X_7_	X_8_	X_9_	X_10_	X_11_	X_12_	X_13_	X_14_	X_15_
γ_0i_	0.201	0.212	0.167	0.172	0.153	0.195	0.201	0.196	0.182	0.154	0.165	0.224	0.249	0.219	0.207

## Data Availability

Not applicable.

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
