# Peer review of "Public Space Satisfaction Evaluation of New Centralized Communities in Urban Fringe Areas—A Study of Suzhou, China"

_ijerph, 2022, doi:10.3390/ijerph20010753_

Round 1
Reviewer 1 Report
General considerations:
The authors present a study on Public Space Satisfaction Evaluation of New Centralized Communities in Urban Fringe Areas – A Study of Suzhou, China. The dataset referred has more than five years, given that it refers to a population satisfaction study, its requirements may have changed over this period.
Abstract should clearly indicate quantitative values to improve the information transmitted, as well as to indicate that the dataset has more than five years (September 2016 to December 2016).
There is only one reference in the last 5 years (2017), all the others are older.
It is suggested that in the data presented, the "local centralization type communities" and "heterogeneous centralization type communities" be identified.
The paper should be carefully revised:
(Line: 60)
It is suggested a reorganization in the sequence of the references (ex: line 60, where it refers [3, 14], put [3,4], changing ref.[14] for ref.[4]...)
(Line: 72-77)
"Considering the available research, this study focuses on the aspects, connotations to [28], and factors [18] of public space satisfaction abroad, while domestic scholars focus on the measurement [13], factors [27], and evaluation models [15] of public space satisfaction."
For a better understanding there should be more specificity in the 'connotations'; 'factors',... etc
(Line: 79)
There should be a reference for Grey Relational Analysis (GRA)
(Line: 86-87)
Add a paragraph to introduce section 2 (between: 2. Method and research objects AND 2.1 Research method. Same in: section 3 (Line: 136-137); section 3.3 (Line: 178-179))
(Line: 87-101)
Include references that support the authors' statements.
(Line: 130)
Clarify which study it is based on for the Centralization Pattern
(Line: 148)
Specify the 4 types referred
(Line: 151-152)
It is suggested that the authors provide evidence on the study conducted that led to the identification of the 15 indices, as well as the elimination of the indices that are difficult to quantify
(Line: 162-163)
The authors refer to 'LiKete's Five Scaling Method', is it not Likert Scale? A reference should also be included
(Line: 167)
The authors referred Figure, but should refer Table
(Line: 177)
In table 3 the attributes 'education level' and 'occupation' must have each of their respective levels explicit.
(Line: 182)
The authors refer table 4 (should be table 2?), because 'Table 4' does not exist in the document (renumber all tables)
(Line: 189)
Again, the authors refer ‘Figure 5’ when they should refer ‘Table 5’
(Line: 193)
Equation (1) is not referred in the text
(Line: 215)
Adjust table 6 content
(Line: 220)
Again, the authors refer ‘Figure 7’ when they should refer ‘Table 7’ (Same in: (Line 265))
(Line: 266 -268)
Uniform data presentation
(287)
Topic 4.3 is poorly developed, and "Inverse correlation" is never done/presented (Same in: (Line: 326))
Author Response
Response to Reviewer 1 Comments
Point 1: It is suggested a reorganization in the sequence of the references (ex: line 60, where it refers [3, 14], put [3,4], changing ref.[14] for ref.[4]...)
Response 1: I've adjusted the order of the papers, which changes ref.[14] for ref.[4]. The order of the literature in the later paper is also adjusted accordingly.
Point 2: (Line: 72-77)"Considering the available research, this study focuses on the aspects, connotations to [28], and factors [18] of public space satisfaction abroad, while domestic scholars focus on the measurement [13], factors [27], and evaluation models [15] of public space satisfaction."
For a better understanding there should be more specificity in the 'connotations'; 'factors',... etc
Response 2: The order of the literature was adjusted and the content on satisfaction factors was added. It’s modified as follows:In terms of research content, the research focuses on the connotation of public space satisfaction[10,11], influencing factors[12,13], evaluation model construction and measurement[14,15], etc. Existing research suggests that “Service Access,” “Social Se-curity,” “Dwelling Record” and “Physical Specifications of Dwelling Place” have a significant relationship with the “Youth Satisfaction” [13].
Point 3:(Line: 79)There should be a reference for Grey Relational Analysis (GRA).
Response 3: The relevant reference has added:
- Deng, J.L. Gray System Basic Method. Huazhong University of Science and Technology Press: Wuhan, Hubei Province, China, 2005, 82-95.
Point 4: (Line: 86-87)Add a paragraph to introduce section 2 (between: 2. Method and research objects AND 2.1 Research method. Same in: section 3 (Line: 136-137); section 3.3 (Line: 178-179)
Response 4:A paragraph has been added between 2. Method and research objects AND 2.1 Research method.Same in: section 3; section 3.3 .
Point 5:(Line: 87-101)Include references that support the authors' statements.
Response 3: The relevant references have added:
- Richard, A.J; Dean, W.W. Translation by Lu X. Applied Multivariate Statistical Analysis. Tsinghua University Press: Bei Jing, China, 2001.
- Li, X.Y.; Hong, Z.S.; Yuan, Y.Q. -et al. Research on Residence Outdoor Space Suitable for Elders and Children's Activities. Urban Development Studies. 2015, 22(5), 104-111.
- Jiang, Z.; Zhang, J.; Qi, Q.Y. The Evaluation on the Visual Satisfaction of Environmental Space of Urban Sculpture-Taking three Environmental Space of Urban Sculpture in Nanjing as an Example. Economic Geography. 2008, 28(6), 1012-1014+1019.
- Murray, D.; Howat, G. The relationships among service quality, value, satisfaction, and future intentions of customers at an Australian sports and leisure centre. Sport Management Review,2002, 5(1), 25-43.
Point 6: (Line: 130)Clarify which study it is based on for the Centralization Pattern
Response 6:Firstly,this study divides new centralized patterns of communities into two types: heterogeneous centralization and local centralization, according to their space scale and causes of formation. Then, it has added related literature :
- Wang, Y.; Feng, B.W.; Li, G.B. Comparison of construction types and operation mechanisms on rural centralized communities—A case study of four communities in Taicang. China Agriculture Resources and Regional Planning. 2016, 37(8), 150–157.
Point 7: (Line: 148)Specify the 4 types referred
Response 7: Based on this, this paper summarizes the satisfaction evaluation index of public space into four categories, which involves satisfaction of space environment, satisfaction of social management, satisfaction of site facilities, and satisfaction of cultural environ-ment.
Point 8: (Line: 151-152)It is suggested that the authors provide evidence on the study conducted that led to the identification of the 15 indices, as well as the elimination of the indices that are difficult to quantify
Response 8: The article further explains the determination process of 15 indicators and how to eliminate the indicators that are difficult to quantify.
We invited 30 experts (including community administrators, community planners and professors) and 80 community residents to participate in the selection of evaluation indexes.Establish a new centralized community public space satisfaction sample database based on formal questionnaires and field interviews.SPSS 21.0 software was used for reliability, validity and exploratory factor analysis, and indicators with Cronbach's α less than 0.7[38] and validity (KMO and Bartlett spherality test) less than 0.5[39] were eliminated, such as pavement form, functional layout and perfection of pointing sign. Finally, from both material and social as-pects, the satisfaction evaluation index system of the new centralized community public space in the urban fringe is determined, which includes 15 indexes in four dimensions, namely space environment, site facilities, social man-agement and cultural environment (Table 2).
Point 9: (Line: 162-163)The authors refer to 'LiKete's Five Scaling Method', is it not Likert Scale? A reference should also be included
Response 9: It has been modified to Likert Scale Method and add the related literature below.
39.Gao, H.H.; Xu, Y.; Gu, X.B.; Lin, X.Y.; Zhu, Q.X. Systematic rationalization approach for multivariate correlated alarms based on interpretive structural modeling and Likert scale. Chinese Journal of Chemical Engineering. 2015, 23(12), 1987-1996.
Point 10: (Line: 167)The authors referred Figure, but should refer Table
Response 10: It has been revised to table.
Point 11: (Line: 177)In table 3 the attributes 'education level' and 'occupation' must have each of their respective levels explicit.
Response 11: The level of education and occupation has been explicited
The average education level is a little low, with 60% of the interviewees’ education levels being lower than an associate degree. 59.2%Most of interviewees make a living as servants,workers or self-employed households, showing that generally, employment levels were generally low.
Point 12: (Line: 182)The authors refer table 4 (should be table 2?), because 'Table 4' does not exist in the document (renumber all tables)
Response 12: It has added Table 4 and renumbered all tables.
Point 13: (Line: 189)Again, the authors refer ‘Figure 5’ when they should refer ‘Table 5’
Response 13: It has changed Figure to Table.
Point 14: (Line: 193) Equation (1) is not referred in the text
Response 14: Equation (1) has been mentioned in the last line of 3.3.2
Point 15: (Line: 215)Adjust table 6 content
Response 15: The contents of Table 6 have been adjusted to correspond to the grade scores of the two different modes
Point 16:(Line: 220)Again, the authors refer ‘Figure 7’ when they should refer ‘Table 7’ (Same in: (Line 265))
Response 16: It has changed Figure to Table.
Point 17:(Line: 266 -268)Uniform data presentation (287)
Response 17: The expression has been modified:
Indexes such as perfection of lighting installation, perfection of sanitation facilities, and brick rate, however, have a relatively smaller influence on public space satisfaction, with values of 0.165, 0.154, 0.153 respectively.
Point 18: Topic 4.3 is poorly developed, and "Inverse correlation" is never done/presented (Same in: (Line: 326))
Response 18: The discussion of "Inverse correlation" has been added in Topic 4.3.
The results shows that the three indexes of accessibility, daylighting and perfection of fitness facilities were located in the "high satisfaction-high significance" quadrant. Activity participatory, activity diversity, activity frequency, cultural landmark richness, management and maintenance of facilities and environment were concentrated in the "low satisfaction-high significance " quadrant. The perfection of resting facilities, vegetation richness, safety, perfection of lighting facilities, perfection of lighting installation and brick rate are in the "high satisfaction- low significance " quadrant. From the perspective of factor distribution of satisfaction in public space, most factors are concentrated in the "low satisfaction-high importance" and "high satisfaction-low importance" quadrants, while few factors are distributed in the "high satisfaction-high importance" quadrant, showing a significant reverse relationship. This also shows that social and cultural factors that are more important to residents' daily life, such as participation, diversity and frequency of activities.As the lack of corresponding attention in the construction and management of new centralized communities, it results in low satisfaction of public space. However, the relatively low level of importance of the brick rate, the perfection of sanitation facilities, lighting installation, etc., has a high degree of satisfaction, indicating that the construction of new centralized community public space in the material level can basically meet the needs of residents at the current stage of life.
Reviewer 2 Report
All is as in the report.

Author Response
Point 1: Let authors check some statistical approaches, the names of all the journals in the list of references including the abbreviated of ones, also check the English in whole text.
Response 1: I have checked some statistical approaches, all the journals in the list of references and the expression of English.
Reviewer 3 Report
The subject is relevant and the paper is concise and well-written.
The methodology is clear and well-presented.
Just one minor remark:
Please consider, if possible, to include references to similar studies concerning public satisfaction and rapid urbanization of urban-rural fringe in other parts of the world.
Author Response
Point 1: Please consider, if possible, to include references to similar studies concerning public satisfaction and rapid urbanization of urban-rural fringe in other parts of the world.
Response 1: The relevant references have added:
21.Saeed, Z.S.; Ali, H.; David, S.; Fatema, H. Fringe more than context: Perceived quality of life in informal settlements in a developing country: The case of Kabul, Afghanistan. Sustainable Cities and Society. 2020, 63, 102494.
22.Wang, X.X.; Shi, R.T.; Wei, W.C. Research on Construction of Public Spaces for Quality Elderly Care Communities in Macao. Journal of Urban Planning and Development. 2022, 148, 04022023.
23.Park, J.Y.; Jeong, J.J.; Park, W.J. A Study on the Community Space that Affect the Public Rental Housing Satisfaction Determinants. KIEAE Journal. 2016, 16, 95-101.
24.John, F.H.; Hugh, S.; Christopher, P. B.L. How happy are your neighbours? Variation in life satisfaction among 1200 Canadian neighbourhoods and communities. Plos One. 2019, 14, e0210091.
25.Wang, C.; Zhang, L.; Ye, Q.L. Public space restructure of new rural community based on peasant household’s satisfaction evaluation—A case study on Dazhu new village in Chongqing Municipality, China. Western Human Settlement. 2016, 31(3), 68–74.[CrossRef]